# An Analysis of Tourism Demand as a Projection from the Destination towards a Sustainable Future: The Case of Trinidad

Lestter Pelegrín Naranjo [1], Norberto Pelegrín Entenza [2,3] and Antonio Vázquez Pérez [4,*]

1   Facultad de Filosofía y Letras, Universidad de Alicante, San Vicente de Raspeig, 03690 Alicante, Spain; lpe14@alu.ua.es
2   Doctoral Program in Tourism, University of Alicante, San Vicente de Raspeig, 03690 Alicante, Spain; norberto.pelegrin@utm.edu.ec
3   Faculty of Administrative and Economic Sciences, Tourism Career, Technical University of Manabí, Ave: Universitaria and Ché Guevara Street, Portoviejo 130105, Manabí, Ecuador
4   Facultad de Ciencias Matemática, Física y Química, Universidad Técnica de Manabí, Ave: Universitaria y calle Ché Guevara, Portoviejo 130105, Manabí, Ecuador
*   Correspondence: antoniov5506@gmail.com; Tel.: +593-995858488

**Abstract:** The objective of this work is to offer an analysis related to tourism demand as a sustainable projection for the tourist destination. The behavior of demand as a key element for subsequent decision-making in management models and strategic proposals for sustainable development centered on the tourist destination constitutes the central element of this work. The study is based on an analysis of tourist activity in the city of Trinidad. In accordance with the quantitative paradigm, an analysis of internal and external secondary information on tourism was carried out based on the results of the surveys, interviews, Likert-type scalograms and semantic differentials completed by the actors involved in the tourist activity in the studied territory. Qualitative research, which is considered a subjective view of the actors involved in the work, was also employed. The main indicators of the tourism demand for the destination and the imbalances that cause dissatisfaction are shown, such as the lack of systematic studies on tourism demand and few coordinated actions between the public and private sectors to satisfy it. It is concluded that the character of Trinidad as a Cultural Heritage of Humanity destination continues to exert an important influence on the demand for tourism in the destination. As a result, it is necessary to delve into new proposals to ensure that tourism in the local context is an economically, socially, and environmentally sustainable activity. For this, the lack of coordination between the state and private sectors must be overcome.

**Keywords:** tourist demand; tourist destination; integrated destination management; Cultural Heritage of Humanity





## 1. Introduction

For tourism, demand includes users whose needs are translated into the consumption and experience of places. Tourists represent purchasing power and the consumption of various tourist products such as souvenirs, clothing, lodging services, food and entertainment [1]. The main motivation for tourism tends to focus on the tourist destination itself, since generally, a tourist wants to become part of the local community that interests them, and share their customs, culture, ways of eating and lifestyle. The consumption of tourist places manifests itself as the subjective experience of the tourist [2,3].

Tourism is not only a means of recreation and the use of leisure time; it is also an instrument to promote interculturality and sustainable economic development, which are much more relevant when carried out in a healthy local environment that encourages the use of indigenous resources, territories, and localities [4,5]. Tourism activity is transformed into a range of possibilities that foster a special market dynamic between tourism supply and demand [6,7].

In the last 60 years, tourism has experienced constant expansion and diversification and has become one of the most important economic sectors worldwide [8,9].

International tourist arrivals went from 25 million in 1950 to 1400 million in 2018. International tourism receipts went from USD 2000 million in 1950 to USD 1,700,000 million in 2018. In 2017, international tourism corresponded to 7% of the world's exports of goods and services, compared to the 6% obtained in 2014 [10].

As a world export category, tourism ranks third, only behind fuels and chemical products and ahead of food and the automotive industry. In some developing countries, tourism is even the first export sector [8].

America, Asia, and the Pacific have registered a growth in international tourism close to 6% [10]. For Cuba, tourism as a source of foreign currency represents the second biggest industry in the economy, with an average annual growth of 11% between 1990 and 2007 and 6% until 2016, which was more in keeping with the global reality [11,12]. This demonstrates the importance of tourism to the country.

After 54 years of rupture, the reestablishment of diplomatic relations between Cuba and the United States of America in 2015 represented an opening of travel possibilities for North American citizens to the Caribbean Island. The new scenario represented a real potential to turn Cuba into a tourist destination par excellence, given its geographical proximity and the curiosity of North American citizens after long years of isolation. At this juncture, Cuba reached, for the first time, 3 million visitors and the occupancy of 65,600 hotel rooms. Meanwhile, the private sector grew, with 16,839 houses for rent and 1700 restaurants. In addition, there was growth in the number of facilities with private management, which stand out mostly because they are high-quality competitive products [6].

In 2017, Cuba was consolidated as an important destination for international tourism with the arrival of 4,700,000 visitors. This figure represents an increase of 6.4% compared to what was expected at the end of 2016 [13]. In 2018, the situation was different, with a decrease in tourist activity of 5.67%, and in the case of American tourists, the drop was drastic, with a reduction of 23.6%. The decline was also significant in all strategic markets for domestic tourism. There was an 8% decrease in tourism from Canada, 4.3% from France, 15.5% from Germany, 8.8% from the United Kingdom, 21% from Italy, and 0.9% from Spain. The number of tourists between 1 January and 30 June 2018 was 2.5 million, which represents a decrease of 5.67% [3,14].

The restrictive measures imposed by the government of the United States of America after the beginning of the presidency of Donald Trump as a continuation of the economic blockade that has maintained the North's power over the country for more than 50 years was aggravated as of 2020 due to the impact of COVID-19. The pandemic strongly affected the flow of tourists on an international scale [15]. This resulted in the authorities of the island anticipating a decrease in the annual number of travelers to 4.7 million [16].

Tourism represents the second greatest source of income for Cuba after professional services abroad, with a contribution of 10% to the gross domestic product (GDP) and the generation of half a million jobs [17].

According to some authors, Cuba's tourist attractions have advantages and are competitive with respect to other Caribbean countries since the country can favor the diversification of its tourist products in the interest of increasing demand and supply. Among its advantages are its greater size and geographical situation, its climatic characteristics, and the greater diversity of its relief. These factors determine the high variety of natural and anthropic landscapes existing on this Caribbean Island. On its coasts, there are important aquatic and terrestrial resources of tourist interest that offer an exuberant natural beauty. Moreover, the country has political and social stability that guarantees security for tourists, with a hospitable population that has a high cultural and health level. Moreover, the labor force has high technical and professional qualifications related to tourist activity. The country has extensive road and airport infrastructure, with an extensive system of technical networks for electricity, communications, and drinking water that cover the entire national territory, as well as adequate environmental legislation. An infrastructure of accommo-

dations, restaurants, and recreational and other services is in full expansion, both in the state and private sectors. The development of higher studies in tourism in a network of universities and an extensive postgraduate plan [18,19] guarantee that Cuba is an important tourist destination, especially for citizens of North America due to its proximity.

The main tourist attraction in Cuba will continue to be its sun and beaches for the next few years. This attraction accounts for 67% of the housing capacity, and in addition to the beach, nautical activities, fishing, diving, and other products are included. Additionally, there is also city tourism, which accounts for 25% of the island's accommodation capacity. In this tourist modality, the patrimonial, historical, and cultural values of many cities in the country are associated, including the city of Trinidad, in which there is the possibility of combining tourism with events, meetings, education, and other areas of interest [19,20].

There are authors and institutions that estimate several scenarios for the development of tourism in Cuba based on the integrated diagnosis of both national and foreign activity [21–23]. The sustained annual growth of half a million visitors between 2017 and 2019 allows us to estimate between 5.2 and 6 million tourists for the next few years [19], a figure that is very close to the scenario forecast by the World Tourism Organization in the year 2000 [24,25].

The objective of this work is to offer an assessment of the tourist demand as a projection for the sustainable future based on the destination in the context of the Trinidad case study, which can constitute a reference for other similar destinations in the country and in the Caribbean area.

## 2. Materials and Methods

### 2.1. Study Area

The study was carried out in the city of Trinidad, which was officially founded as Villa de la Santísima Trinidad and called by some historians and inhabitants "La Trinidad" or "Trinidad de Cuba". It is a city located in the central region of the country, specifically in the south of the Sancti Spíritus province, and is the capital of the municipality of the same name. In 1988, the United Nations Educational, Scientific and Cultural Organization (UNESCO) inscribed the historic center of Trinidad along with the Valle de los Ingenios on the World Heritage List since it is an area where sugar production traditionally prospered during the 19th century. This fact makes Trinidad an important place of tourist interest for Cuba.

The city of Trinidad is located in the tourist region of the south-central coast of Cuba. It was founded at the beginning of the year 1514 by the colonizer Diego Velázquez y Cuéllar. It has existed for more than five centuries, for which it qualifies as one of the oldest and best-preserved colonial cities in the country and in Latin America [26,27]. In the public documents of the colony from the year 1587, it was already recognized as a city. The historic center is distinguished as a World Heritage Site. It stands out because it has been inhabited since its foundation, and residential use has marked most of its buildings [28–30].

Figure 1 shows the map with the location of the city of Trinidad and the location of the main tourist activations.

The initial economic activities were soon replaced by the production of cane sugar, taking advantage of the soils of what is now called Valle de los Ingenios.

By the end of the 18th century, Valle de los Ingenios had 90 cane sugar factories that produced between 50 and 60 thousand arrobas of sugar and about 700 jars of honey in each harvest. Thousands of slaves worked from sunup to sundown in the sweet grass plantations and in the mills to make sugar [31].

The favorable economic conditions attracted emigrants from various parts of the world. Traces of exquisite Italian painters, expert Catalan master builders, and blacksmiths forging bells and cast-iron grills remained in Trinidad, incorporating the fruit of their labor into city residences and the no less opulent mansions for recreation that the owners had built near Trinidad and their cane sugar-producing mills [28,31]. The city prospered and grew. In 1827, 12,543 inhabitants were registered in the urban area of the city [32].

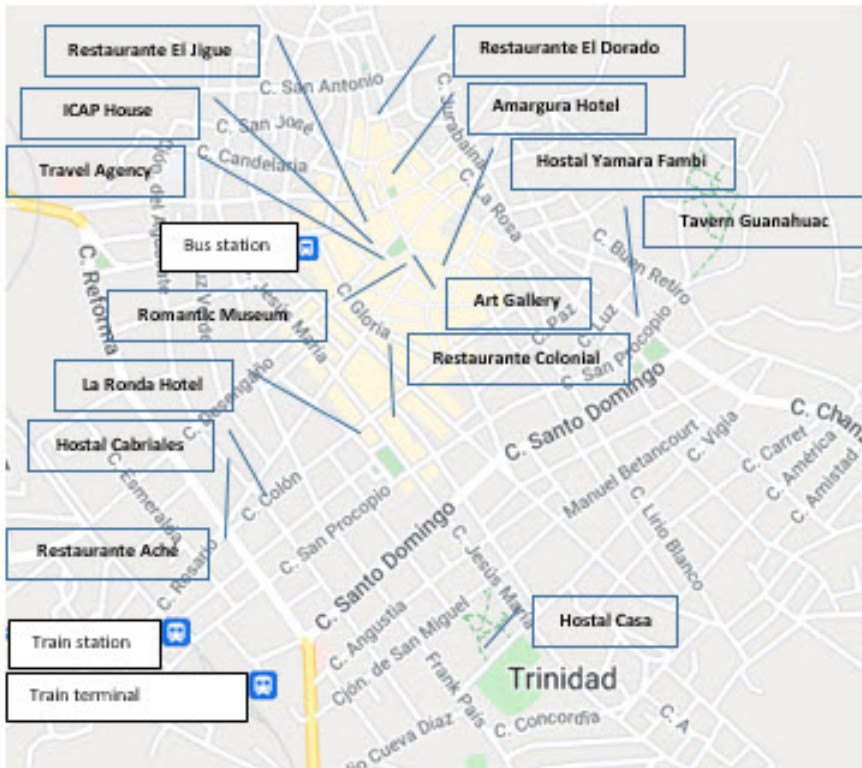

**Figure 1.** Location of the tourism activations studied. Source: own elaboration from Google Earth.

The city of Trinidad is an urban space that, for more than half a millennium, has witnessed successive periods of decadence, misery and splendor, which is currently shown to the visitor as a real marvelous heritage setting. In Trinidad, one can witness the manifestations of the heterogeneity that has characterized this city, ranging from the secrets and traditions inherited from the ethnic waves that successively contributed to its population and expansion, to the rich exponents of art, goldsmithing and opulence that were accumulated and protected from looting and prying eyes by its inhabitants for decades. These elements are now brought to light in the city, endowing its ancient buildings and cobbled streets with growing attraction [27].

The city began to show off large mansions, squares, and plazas for the leisure and social recreation of its residents. Streetlights were placed for public lighting and the streets were covered with stones that give the city a peculiar configuration of urban traffic. However, the splendor was doomed to fade. The city's economic infrastructure had two weaknesses: a production system based on plantation cultivation by slaves, which was stopped upon the development of the productive forces of the already thriving capitalist production system, and the limited physical space of the valley, whose overexploited soils and decrease in forests that provided energy resources began to have a negative impact on agricultural and industrial yields at a time when sugar production in Europe began to displace the Cuban sugar in the market [31,33].

As it was impossible to stop the crisis even with the introduction of technological improvements, capital emigrated to other areas of the country, and from the 1940s, the former city began to show obvious signs of decline and economic backwardness. The masters abandon their city mansions, recreational hacienda houses, and properties, leaving the care of the riches contained in them in other hands, perhaps with the aspiration of returning in new times of prosperity, times that would never be repeated again [26].

The Association Pro Trinidad was founded in 1941 and was assumed as a commitment to turn the city into a garden and promote tourism within it. However, without effective support from government actors, this initiative failed, and the city remained ostracized until the advent of the 21st century, in which the Cuban government authorities accepted the

irruption of tourism as a necessity for the economic and social development of the country because the economic dividends it generates are capable of promoting the reactivation of the other sectors of society. As a consequence, this brought about the revival of the city and its sites of heritage interest, as well as the creation of the infrastructure that would allow for assimilating the arrival of a growing number of visitors [27,34].

Hence, there is significant importance in addressing the characteristics of the demand for tourism in the city of Trinidad [35], as it is a key element for the sustainability of socio-economic development, which is one of the determining factors in management models and the proposed strategies for progress in the city. From a synthesizing perspective, Trinidad is a heritage tourist destination, which covers not only the city but also the nearby Valle de los Ingenios and the surrounding areas of the big park Sierra del Escambray [28,36].

Before the COVID-19 pandemic, the city of Trinidad was considered one of the most in-demand spaces by international tourism [16]. In this regard, it should be noted that some authors prefer the term 'tourist space' rather than the commonly used terms 'area' or 'region' when referring to the physical geographic environment that contains the activities usually carried out by tourists [37].

### 2.2. Methodological Framework

To contribute to the achievement of the proposed research objective, a descriptive study aimed at the analysis and description of the current situation of the city of Trinidad was projected on the basis of a mixed paradigm (linking qualitative and the quantitative approaches) in order to identify the potential for tourism demand [38]. Figure 2 shows the diagram of the investigation procedure.

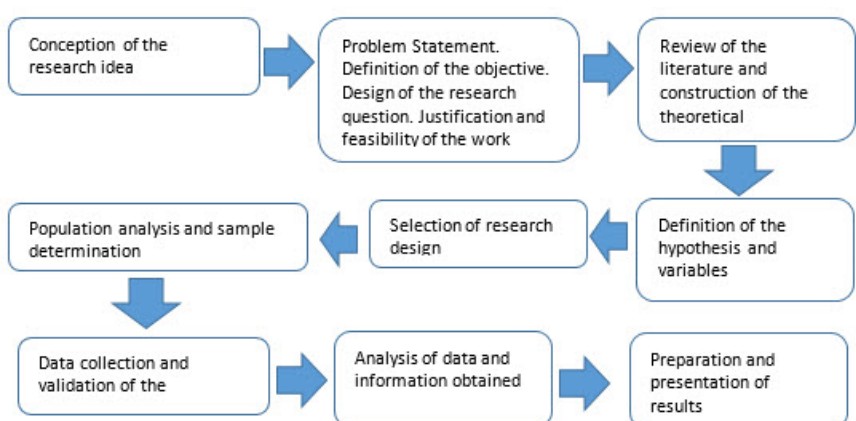

**Figure 2.** Diagram of the investigation procedure.

The deductive method, observations, and field work were applied, which allowed us to conduct an analysis of the problem and reach precise conclusions on the subject under study. This made it possible to carry out consultations with specialists on the subject of tourism, as well as to form discussion groups with actors related to tourism in the city. The documentary analysis technique was applied.

Some institutions of the territory participated in the discussion groups, such as the Ministry of Tourism (MINTUR), the Ministry of Agriculture (MINAGRI), Flora and Fauna, the Provincial Directorate of Physical Planning, and travel agencies. Their input was sought to identify and evaluate conceptual aspects and methodological and practical experiences related to the research.

The research started with an investigation of the background and elements that have contributed to the tourist boom that is currently being experienced by the destination of Trinidad de Cuba. To this end, the concepts of the demand and management of local development are deepened, as well as the close relationship between the two. The elements of Trinidad's natural and cultural heritage are addressed based on the various modalities that

have led to stable tourist demand. Various approaches to the subject are analyzed, emphasizing innovation, the emergence of new trends, revitalization, the adequate management of the poles, the relationships established between the state and private sectors, as well as the design of tourism products as a necessity for post-COVID-19 tourism management [14].

A mixed approach was followed, with a combination of quantitative and qualitative elements. Moreover, through the deductive process, limited approaches were conducted using statistical data, upon which we tried to base our study on the demand for tourism, which allows for a greater breadth and richness in the interpretation of the data results.

With these elements, the conditions that currently characterize the tourism demand in the heritage destination of Trinidad and the Valle de los Ingenios are proposed in correspondence with the diversity of the models that involve the tourism actors, both from the state and private sectors, and their repercussions for social development.

### 2.3. Data and Data Sources

The internal primary information was obtained from the review of the balance reports of the tourist activity in the destination of Trinidad provided by the various facilities and chains about the entry of tourists, the behavior of the levels of occupation, and financial balances. In accordance with the quantitative paradigm, the analysis of the statistical data derived from the economic indicators of efficiency by the superior organs of economic direction (OSDEs) of the tourist entities in the Trinidad destination was carried out, as well as the reports of the balance of the work carried out in the previous period, which includes the efficiency indicators of the main entities. Pertinent comparisons were made. The data provided by the regulatory entities of the non-state sector were also analyzed.

The investment and land-use plans of the municipality, the government development strategies, the work balances of the Municipal Directorate of Labor and Social Security, as well as the results of the National Tax Office (ONAT) were reviewed and assessed.

The qualitative paradigm was also considered throughout the entire investigative process, which considers the subjective views of the actors involved in the process to be essential. Their assessments do not always coincide with the cold statistical analysis from official figures, as occurred with the lessors of houses, rooms, and spaces, as well as other actors linked to tourism services in an unofficial way.

The internal quantitative data were obtained through the application of a survey. The qualitative information was reinforced with interviews with key informants linked in one way or another with the tourist activity in the territory. Semi-structured interviews were carried out with informants on thematic areas related to the various tourist modalities to determine the characteristics of the relationships that have been established. A Likert-type scalogram was presented to various actors with incidence to evaluate their attitude towards the established practices.

In all cases, a non-probabilistic sample of 50 voluntary subjects was selected, all of whom agreed to provide information on the aspects included in the surveys and who met the requirements demanded by the objectives pursued in each of the instruments. The subjects surveyed were divided into two subgroups of 25 subjects, each corresponding to their employment affiliation to either the state or private sector. The instruments were applied throughout the year 2019, and it was possible to obtain information from 150 subjects, 75 of whom were from the state sector and 75 of whom were from the private sector.

The external primary information was obtained by applying a Likert scalogram to 50 members of the existing communities in the various tourist destinations and products (Heritage City, Peninsula, Valle de los Ingenios, and Gran Parque Guamuhaya) to determine the effects of tourism. In addition, various instruments were applied to several non-probabilistic samples of tourists who agreed to answer the surveys (voluntary subjects). There were 100 in each of the instruments at different times throughout the year 2018.

In all cases, the instruments applied to tourists were made in Spanish, English, and French to achieve greater ease in their application and processing, and an item present in all of the surveys was the country of origin.



The instruments included a semantic differential on the tourist products offered in the destination and a questionnaire about the modalities of their interest in the destination of Trinidad. In addition, the reasons why tourists selected the destination of Trinidad during their visit to Cuba, the repetition of visits to the destination that indicates fidelity as a selection criterion, and the measurement of the level of satisfaction of the clients with respect to the tourist products offered were included in the questionnaire. Moreover, based on the criteria they expressed in the survey and what they had to do to meet these criteria, the respondents were asked to evaluate two essential aspects: the level of sacrifice, understood as the perception of the effort they had to make to access the different tourism products in Trinidad, and their assessment of the magnitude of the benefit received. The aspects were valued in the high and modest categories, respectively.

A total of 300 clients were surveyed throughout the second half of 2019 (100 in each instrument), 150 of whom were staying in hotels and tourist facilities belonging to the state sector and 150 of whom were staying in hostels operated by private entrepreneurs.

The information was also obtained through the data analysis of many reports, information materials, outreach brochures, web pages, and other materials that provided data on the creation, management, promotion, and marketing of the Trinidadian tourism product.

There is no doubt that the threat or probability of suffering from the effects of a disease, the fear of being caught in the middle of a conflict, or other reasons that affect personal security will influence the motivations to travel to a destination and will influence the decisions to be made.

These three approaches coexist in the results of the analysis of tourism demand, and their points of view are totally useful if they are considered integrally. To carry out successful commercialization, a multidisciplinary analysis is required. Some authors [39,40] have recognized the existence of two large groups of models related to demand: the explanatory or determining factors in the travel purchase process and the tourist dynamics of tourist flows or motivational factors that influence the characterization of destinations and forecasts.

### 2.4. Theoretical Delimitation

Some authors [41] argued that the correct management of tourist destinations depends, to a large extent, on the forecast of customer demand [42–44]. The most recent studies on the behavior of tourism demand in Cuba have been carried out by González, Santa Cruz, Díaz, and others [45,46].

Tourism demand models start from their own conceptual definition. They arise from the people who have the three minimum essential conditions to travel (free time, personal income, and an interest or need to make the trip) and the decision they adopt that tempers their own particularities, possibilities, and interests [9,25,47].

Although it is true that income constitutes a basic factor in the analysis of demand, it is also true that the temporary restrictions that are determined both by the working conditions of the people, as well as family restrictions themselves [48,49], have a notable influence when making decisions related to travel. This can be seen in practice with the conditions imposed by the COVID-19 pandemic [15].

For the authors, demand depends on various factors related to unconventional variables, such as the occurrence of a significant event in each scenario, be it natural, economic, social, or environmental [50]. The authors of [51,52] agree on this point, as they have stated that the consumption of tourist places manifests itself as a subjective experience of the tourist. In addition, [53,54] have referred to the Cuban case as a country with very particular socioeconomic conditions given the direct impact of the economic and commercial blockade of the United States on the island.

The demand can also be approached from the economic point of view to consider the volumes of trips or tourist services for which people are willing to pay a specific price at a given time. Demand can also be approached from the geographic perspective, particularly from the point of view of human geography, where demand is considered as

the current and potential tourist flows themselves. There is also the psychological and sociological point of view, which looks at the motivation and social behavior in a sustainable environment [55–57].

In the first case, income plays an essential role, influencing both personal and family decisions when making the decision to travel. However, the scenario for the enjoyment of leisure time and the conditions determined by the geographical characteristics of the destination can be a decisive factor in decision making [58]. When the holidays are approaching, different media tend to pay special attention to weather information if it is a short period outside the most stable season [59].

A more active cyclonic season than normal, as happened in the tropical Atlantic basin in 2005, influences the decrease in the choice of the Caribbean and Cuba as destinations. An adequate marketing strategy around environmental values and the conservation of nature in each place positively influence the decision to choose it as a destination. The environmental issue becomes a significant issue in the management of tourism activity and especially tourism in natural spaces [60].

## 3. Results

The knowledge of tourism demand is one of the success factors for tourist destinations. It is most common for the entities in the destination city, as is the case of Trinidad, to carry out the study of the demand for tourism to prepare the products that they can offer. However, in the study carried out on the 15 selected tourist activities shown in Table 1, the real situation does not behave in the same way. For this, it was considered to select the activities that present the greatest demand by tourists, in this case, six related to food sales and six to accommodation, in addition to two cultural and one related to tourism services.

**Table 1.** Tourist activities selected to evaluate the studies on tourism demand.

| No | Activities | Service Provided | Resource Management |
|----|-----------|------------------|---------------------|
| 1 | Restaurant Jigüe | Restaurant | State |
| 2 | Restaurant Colonial | Restaurant | State |
| 3 | Guanahuac Tavern | Restaurant | State |
| 4 | Aché | Restaurant | Private |
| 5 | The Sailor | Restaurant | Private |
| 6 | The Golden | Restaurant | Private |
| 7 | Hotel La Ronda | Accommodation | State |
| 8 | Hostel Bitterness | Accommodation | State |
| 9 | ICAP House | Accommodation | State |
| 10 | Hostal Yanara Fambi | Accommodation | private |
| 11 | Hostal Casa Mía | Accommodation | private |
| 12 | Hostal Cabriales | Accommodation | private |
| 13 | Romantic Museum | Cultural | State |
| 14 | Art Gallery | Cultural | State |
| 15 | Travel agency | Services | State |

In the case of one hundred percent of all the activities, it was stated that the information resources on the demand are managed via employing public and private agents. However, in the case of the nine entities of the state sector, several members of their respective boards of directors were interviewed and, in the case of the six private entities, the owner of the enterprise or his or her substitute was interviewed.

The interviewees generally expressed a lack of knowledge about the type of secondary and primary information that is collected, with the lack of knowledge being much more accentuated in private enterprises. Among private enterprises, 66% of the respondents expressed ignorance on the subject, while this figure was 33% for the state entities.

It is useful for any facility to know the issuing markets. This secondary information is very easy to obtain in the tourist studies carried out in the territory, as shown in Figure 3.

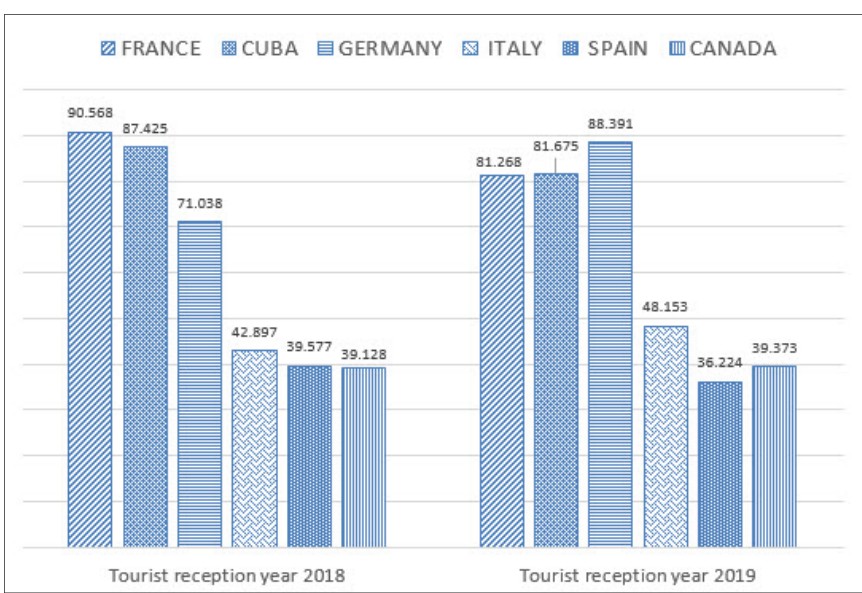

**Figure 3.** Main issuing markets to the tourist destination of Trinidad. Source: own elaboration from [10,11].

Between 2018 and 2019, a slight reduction in the reception of tourists from France, Spain, and national tourism can be seen. Germany, Italy, and Canada show increases in the emission of tourists. In general, there was a slight increase in the tourist reception of the destinations analyzed above, which amounted to 4.451 tourists.

The fact that national tourism has grown modifies the demands of tourists, so differences can be observed with respect to the behavior of these indicators on a national scale, as can be seen in Figure 4. This figure shows the result of a survey randomly applied to 300 tourists staying in various facilities in Trinidad, both from the state and private sectors.

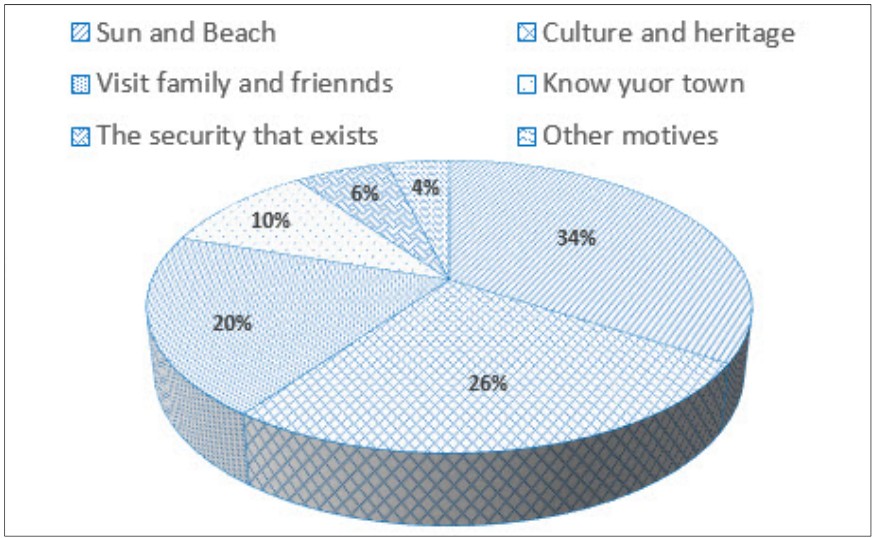

**Figure 4.** Main motives of destiny.

Loyalty is one of the criteria that influences the decision of tourists to prefer to visit the chosen destination, which is manifested through the rate of repetition or reiteration with which the destination is selected by tourists. In the case of Trinidad, the results of the surveys show that 17% of tourists stated that they visited Trinidad more than once, as reflected in Figure 5.

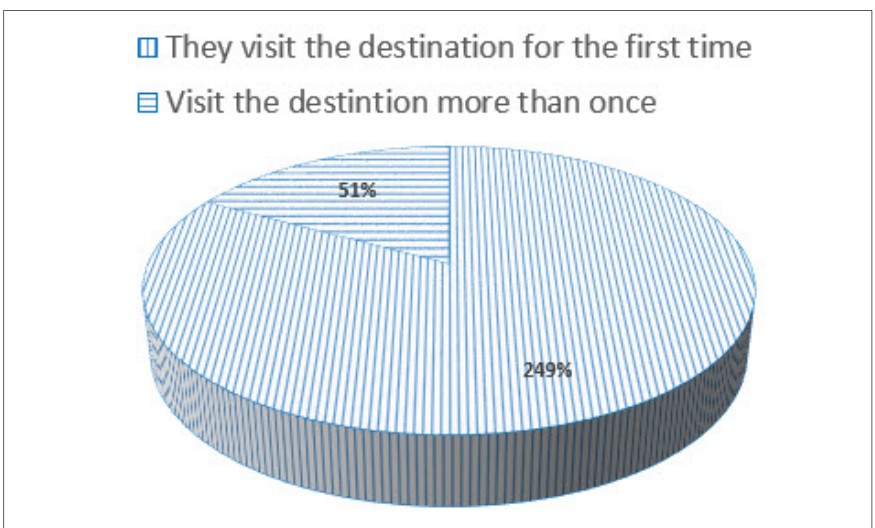

**Figure 5.** Tourists who visited the destination more than once.

The decision-making process is reinforced by the marketing work carried out by tour operators, travel agencies, and individuals themselves to promote their private enterprises. The results of the study of the demand of tourists both national and foreign to the Cubatur Trinidad travel agency in 2018 yielded the results that are observed in Figure 6.

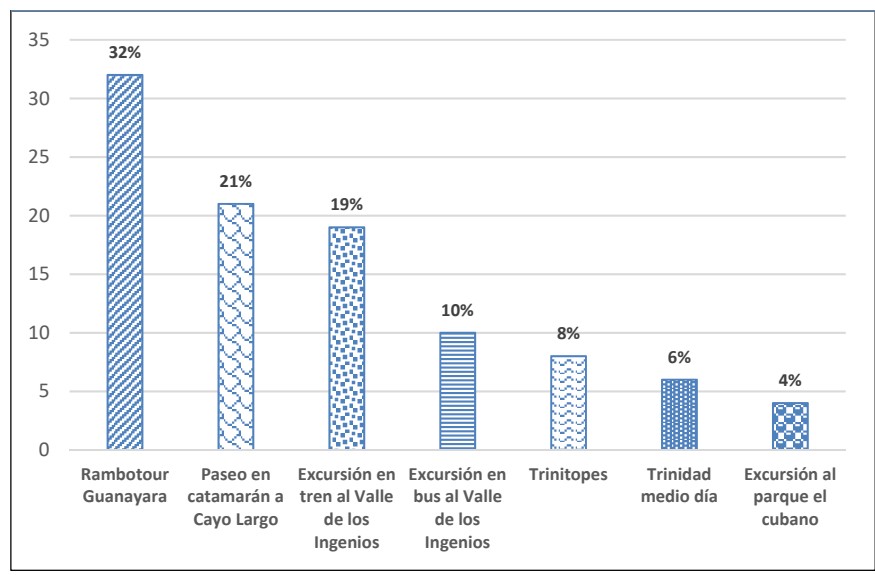

**Figure 6.** Demand made by tourists to the Cubatur Trinidad travel agency.

The numbers of tourists who visited the museums of the city's historic center in 2018 are significant, as shown in the results of the survey reflected in Figure 7.

There is also a great demand from tourists for other activities related to culture that have nothing to do with the built heritage. They are those related to what is called people-to-people contact and for which the city of Trinidad stands out. The shared enjoyment comes from not only listening to but also from the interpretation of the music and the dances of the Casa de la Trova or the Palenque de los Congos Reales.

As a result of the surveys applied to tourists during the development of the investigation, it was possible to determine the most demanded products and which of them were more difficult to obtain by tourists, as shown in Table 2.

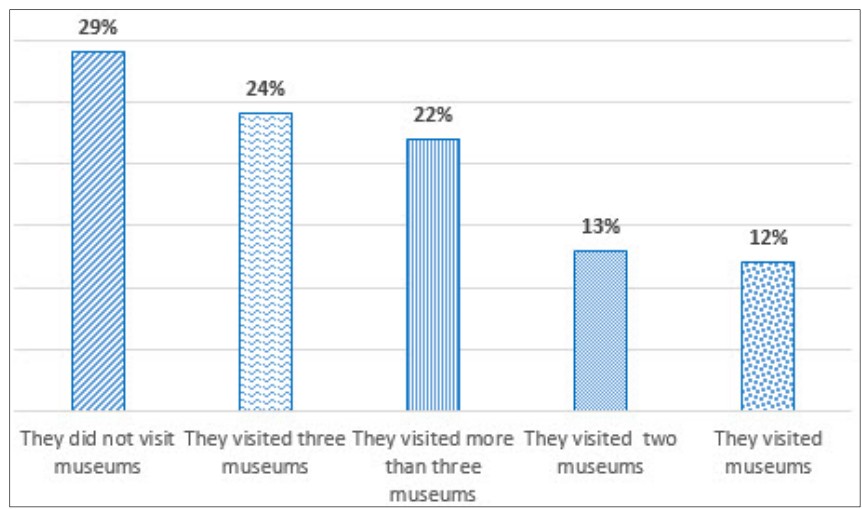

**Figure 7.** Visits made by the surveyed tourists to museums in Trinidad.

**Table 2.** Products demanded by tourists and unsatisfied demands.

| Claims Made | Satisfied Demands | | Unsatisfied Demands | |
|---|---|---|---|---|
| | Tourists in State Facilities | Tourists in Private Hostels | Tourists in State Facilities | Tourists in Private Hostels |
| Enjoy the sun and the beach | 75 | 68 | 0 | 0 |
| Horseback riding | 5 | 50 | 10 | 0 |
| Speleotourism | 3 | 7 | 5 | 0 |
| Gastronomic tourism | 12 | 22 | 0 | 0 |
| Cycle tourism | 7 | 32 | 0 | 4 |
| Agrotourism | 5 | 7 | 3 | 0 |
| Get to know the people and their customs and traditions | 26 | 36 | 7 | 0 |
| Museum visits | 32 | 25 | 0 | 0 |
| Visit to other heritage sites (churches, squares, buildings, etc.) | 35 | 29 | 9 | 0 |
| Observation or photo hunting of species of flora and fauna | 0 | 15 | 13 | 3 |
| Hiking, walking, orienteering tour in the countryside | 5 | 28 | 17 | 7 |
| Enjoy cultural and sports shows | 60 | 45 | 10 | 13 |
| Adventure tourism: climbing, canyoning, rafting, canopy or zip line | 4 | 18 | 10 | 5 |

**Table 2.** *Cont.*

| Claims Made | Satisfied Demands | | Unsatisfied Demands | |
| --- | --- | --- | --- | --- |
| | Tourists in State Facilities | Tourists in Private Hostels | Tourists in State Facilities | Tourists in Private Hostels |
| Sailing and other nautical activities, diving, sport fishing | 25 | 18 | 5 | 5 |
| Academic, learning, and event tourism | 0 | 3 | 0 | 1 |
| Shopping tourism | 16 | 25 | 4 | 0 |

The information reflected in Table 2 allowed us to notice that the demands made by tourists housed in private sector facilities exceeded those made by those housed in state facilities by more than 100. This may indicate that the latter were subject to the offers of a preconceived hotel industry, where the options provided by the facility are already included and the tourists prefer to take advantage of those options without demanding other products. In the case of those who stayed in private-sector facilities, they had the possibility of selecting a group of more diverse tourist products and greater options.

Secondly, tourists staying in state-sector facilities had a higher rate of unsatisfied requests made, which is almost triple that of the same indicator achieved by those staying in private hostels. This indicates that the latter service turned out to be more personalized and provided more attention to the satisfaction of the demands made by their tenants, as shown in the information in Figure 8.

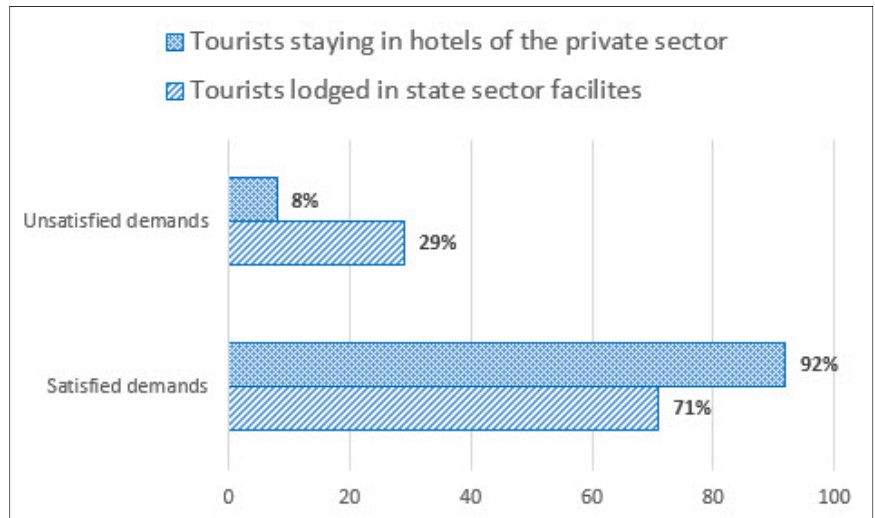

**Figure 8.** Differences in the satisfaction of demand according to the sector.

## 4. Discussion

Qualitatively, the results of the investigation allowed us to verify that the studies carried out on tourism demand are scarce and shallow, as they respond to empirical methods rather than to systematic and properly structured studies. This is because 60% of the managers recognized that systematic studies are not carried out on this aspect in their facilities.

Above all, in private enterprises, there is little knowledge of the origin of the tourists who visit them, especially in those dedicated to restoration, where the opinion is that all those who need the service are served regardless of where they come from. This procedure constitutes a barrier to the provision of a personalized service, as required by most tourists today.

One of the factors that has determined the increase in demand in the destination of Trinidad is the stability maintained in the prices, which, although they have increased, have remained relatively stable in recent years and are comparatively cheaper in relation to other touristic destinations.

It was learned that there is a network of providers within the same city that can be called at any time to satisfy a demand, including taking tourists on a bike ride to the beach, going on a horseback ride through the countryside, and exploring the caves near the city. These providers adjust the price of their services with the landlord that hosts the tourists, and the tourists thus obtain the service they want—something that state entities are prevented from doing because they do not have it included in their plans.

In the quantitative order, the results of the work show a differentiation between the lodging and restaurant services of the state sector, with the former having greater clarity regarding the existing demand. However, they stated that the entirety of the secondary information reaches them through their respective companies and higher business management organizations (OSDEs), but they also apply questionnaires and use other instruments such as tourist opinion records and primary controls. This allows them to have a deeper understanding of the demand in relation to entrepreneurs in the non-state sector.

Domestic tourism maintains an upward trend. However, there is no complete understanding that, for the destination, the national demand must be the most significant and the one that must be attended to with priority. In interviews with people linked to tourism, especially from the state sector, some negative opinions were expressed about national groups, such as that they are not good customers, they leave very few tips, and they complain a lot.

The demand for horseback riding through the Valle de los Ingenios is one of the options that has not been satisfied by state entities, which shows inefficiencies in this sector to act in accordance with the interests of tourists. Tourists prefer to leave the city and get in touch with nature, which leads receiving entities to rethink their offers to satisfy these demands.

In the case of national tourists, visits to museums are significant, since the elements and manifestations that constitute cultural heritage are the results of the ways in which a society or culture relates to its environment [61,62].

Tourist activity can be sustainable if an adequate projection and management of tourist destinations is carried out. When there is no good competitiveness in the tourism sector, it is not possible to achieve the sustainability of the activity. Especially in the case of Trinidad de Cuba, the imbalances between state and private facilities affect the diversification of the tourist product; this is manifested in the dissatisfaction that tourists show when visiting the country's facilities.

Another important element is to achieve a harmonious relationship between social, economic, and environmental interests so that tourism is an activity with a greater projection than it currently has and so that the sector becomes an important element of development in the country.

Nature tourism combined with other tourism products, such as city tourism and sun and beach tourism, can play an important role in generating value in rural areas, where the greatest poverty and social backwardness tend to be concentrated; however, this requires an environmental policy and regulation appropriate to the specific conditions of each territory, capable of being observed by state and private facilities.

The global environmental debate has become more latent since the 1970s in the last century. The member countries of the Organization for the United Nations (UN) in the Conference on Environment and Development [63–65] met in 1972 in Stockholm Sweden, where the concept of development was reviewed in its multiple dimensions, especially the environmental, social, and economic ones.

Sustainable development requires an integral and necessary relationship between the natural system and development, a situation that must be permanently manifested in tourist activity. It refers to a process of change in which the use of natural resources, the

direction of tourist activity, and scientific and technological progress, under the direction of the country, allow the satisfaction of the requirements of tourists without affecting the right of future generations to satisfy their material and spiritual needs in a healthy environment [66].

## 5. Conclusions

In order to achieve competitiveness in the tourism sector, tourist providers in the destination must follow a strategy that pays special attention to a solution to the imbalances derived from the dissatisfaction of tourists, which are accentuated in the facilities operated by the state sector. This situation is of special interest in the post-COVID-19 context.

There are few systematic studies in the local context that are related to the sustainability of tourism demand in the state sector and especially in the private sector. Actions are carried out in a way that is dictated by spontaneity, without the application of technical methodologies. Technical methodologies reveal new knowledge aimed at transforming weaknesses in order to achieve sustainable satisfaction with tourist activity.

There are few coordinated actions between the public and private sectors to arrange actions aimed at the greater satisfaction of tourism demand, which has generated dissatisfaction among visitors.

This investigation allowed us to determine that an unjustified differentiation persists in terms of attention to international and national tourism. Although the demand for tourism continues to be mostly from foreign visitors, national tourism has grown in recent years and now occupies a significant place. However, tourist providers have not met this demand with corresponding actions that would achieve a higher level of satisfaction among national clients.

Tourists from the United States, who constitute the largest and main potential for demand, cannot be considered as a projection of real demand due to the socio-political limitations and regulations imposed by successive US administrations, which expressly prohibit their citizens from traveling to Cuba.

Although Trinidad's status as a Cultural Heritage of Humanity destination influences the preference of tourists, demand has increased due to other offers, such as sun and beach tourism, which continues to be the most in-demand attraction, as well as family visits. Moreover, related options are growing with rural and nature tourism. With the proper management of demand in the local context and solid coordination between the state and private sectors, the greater satisfaction of customer preferences can be achieved, based on the potential that the territory possesses for the various tourism products that it offers.

There are natural and social conditions that can have a better use for proposing new marketing strategies that would allow for increasing the demand for Trinidad as a destination. These include incorporating new elements to the offer and promoting others, such as crafts, cave tourism, and rural tourism, which are insufficiently attended to by the state sector and to a greater extent by the private sector. It is necessary to articulate a proactive strategy that develops concerted actions between both sectors to provide sustainable responses to the increase in and diversification of the demand for tourism.

**Author Contributions:** Conceptualization, L.P.N. and A.V.P.; methodolog A.V.P. and L.P.N.; formal analysis L.P.N. and A.V.P.; investigation L.P.N. and N.P.E.; data curation, L.P.N. and A.V.P.; writing—original draft preparation, N.P.E. and A.V.P.; writing—review and editing A.V.P.; visualization L.P.N. and A.V.P.; supervise N.P.E.; project administration, N.P.E. All authors have read and agreed to the published version of the manuscript.

**Funding:** This research was self-financed with the researchers' own resources.

**Institutional Review Board Statement:** Not applicable.

**Informed Consent Statement:** Not applicable.

**Data Availability Statement:** Data can be provided upon request from the corresponding author.

**Acknowledgments:** The authors gratefully acknowledge the authorities of the Ministry of Tourism and the owners of private tourist facilities in the city of Trinidad and Valle de los Ingenios.

**Conflicts of Interest:** The authors declare no conflict of interest.

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
