# Peer review of "An Analysis of Tourism Demand as a Projection from the Destination towards a Sustainable Future: The Case of Trinidad"

_sustainability, doi:10.3390/su14095639_

Round 1
Reviewer 1 Report
Dear Authors,
After reviewing article „ Analysis of tourism demand as a projection towards a sustainable future from the destination. Trinidad case study” , I have comments and suggestions as follows:
INTRODDUCTION: In my opinion, in paper missed a clear definition of the research goal (in particular in the context of "sustainable tourist projection for this tourist destination"). Moreover, there is no definition of what exactly results brings to research?
STUDY AREA: I suggest placing the location of the research area on the map and 15 selected tourist activities
METHODOLOGICAL FRAMEWORK: This part of the work needs rebuilding. There is no presentation of the research stages used. I propose to include a diagram of the research procedure, which would improve the reader's perception of the work.
DATA AND DATA SOURCES: This part of the work needs to be systematized, first of all I propose to organize the information in quantitative and qualitative terms (line 221)
THEORETICAL DELIMITATION: Change the name of the subsection from Spanish to English (line 288). The content of the work covering lines 331-338 should be transferred to subchapter METHODOLOGICAL FRAMEWORK or DATA AND DATA SOURCES.
RESULTS:
- No selection criteria for tourist activities (line 343).
- No consistency in the data presented 2017 or 2018 or 2019 ? (for example fig. 1 line 358)
- What do the authors mean by Tourist in fig 1 as measured ? (line 358)
- Even on the basis of the data in fig.1 it is worth showing the dynamics of change
- Lack of characterisation in terms of the size of the tourist activities for the analysis (Table 1)
- The perception of research results would be improved by discussing them in a clear division into quantitative and qualitative terms.
- The authors rely on absolute data and why statistical indicators were not used.
DISSCUSION: No discussion of how the authors understand projection towards a sustainable future from the destination ?
CONCLUSIONS: After the amendments have been introduced, I propose to redact the conclusions in this form are too general.
Reviewer
Author Response
"Por favor vea el archivo adjunto".

Reviewer 2 Report
Thank you for the opportunity of reading and reviewing your interesting manuscript. The paper presents the results of a field research, survey-based, and it is well written, although revisions can be made.
My suggestions for improving the quality and scientific soundness of the paper are as follows:
1.try to address the issue of representativeness for the sample. It is not clear why it is representative or, if not, clarify the validity of results
2.a more coherent presentation and a more in-depth and detailed analysis of the results. For now there is only a descriptive presentation of the responses to the survey questions. You should try some statistical treatment of data, correlations between variables etc.
3.in this regards, you should consider formulating several hypotheses or at last some research questions to address; thus your paper will become more like a scientific article than an informative paper
4.in the final section you should refer to the limitations of your research and how they affect the validity of your results. Moreover, practical and theoretical implications should be emphasized.
Good luck!
Author Response
1. Try to address the issue of the representativeness of the sample. It is not clear why it is representative or, if not, to clarify the validity of the results
Added clarification on the representativeness of the sample:
Given the instability of tourist arrivals in the Trinidad destination, it was not possible to define a study population, so we tried to work with an open sample made up of the largest possible number of tourists staying in the Heritage City, Peninsula, Valle de Los Ingenios and Gran Parque Guamuhaya, so that it constituted a representative quantity for the moment of carrying out the research work.
2. A more coherent presentation and a deeper and more detailed analysis of the results. For now, there is only a descriptive presentation of the answers to the survey questions. You should try some statistical treatment of the data, correlations between variables, etc.
3. In this sense, you should consider formulating several hypotheses or finally some research questions to address; therefore, your article will look more like a scientific article than an informational article.
New content was incorporated into the work to provide depth to the results. Some premises and research questions were incorporated that contribute to a better understanding of the information reflected in the work.
4. In the final section you should refer to the limitations of your research and how they affect the validity of your results. In addition, the practical and theoretical implications should be emphasized.
The limitations of the research were incorporated, as well as the practical and theoretical implications:
Within the methodological limitations of the research, it can be pointed out that a finite population could not be defined for the study, given the instability in the arrival of tourists in the Trinidad destination. This situation involved working with a random sample, trying to achieve the greatest possible representativeness, which is very difficult to achieve under these conditions. Another limitation is related to the lack of previous studies related to the projection of demand in Trinidad, which made it difficult to process comparative data with other periods. However, it can be considered that the information obtained for this study is valid, which will allow its comparison with other investigations that are carried out, related to the subject studied.
Within the practical implications, it can be pointed out that the results of the research can help to improve the tourist demand in the Trinidad destination as a projection towards a sustainable future, especially in the need to achieve adequate coordination between the facilities. of the state and private sector, as well as achieving greater use of the potential offered by the natural environment, provided that it is carried out in compliance with the requirements established in the legislation and the country's international commitments on environmental issues.
Within the theoretical implications, it can be pointed out that the research focused on offering a group of research inputs linked to the projection of tourism demand in the Trinidad destination as a projection for a sustainable future, taking as principles open-ness, communicability, reflexivity, and data analysis to reach a qualitative result that serves as a starting point for the development of other research related to the subject studied.
All of this allowed the conceptual and practical analysis associated with the projection of the tourist demand in the Trinidad destination, to define the research questions and the establishment of a group of paradigms that allowed to fulfill the objective of the investigation and reach precise conclusions. on the subject studied.

Round 2
Reviewer 1 Report
Dear Authors,
I was pleased to revisit yours article. I propose to accept it in present form.
Reviewer